# Generalized Correspondence-LDA Models (GC-LDA) for Identifying Functional Regions in the Brain

**Timothy N. Rubin**
SurveyMonkey

**Oluwasanmi Koyejo**
Univ. of Illinois, Urbana-Champaign

**Michael N. Jones**
Indiana University

**Tal Yarkoni**
University of Texas at Austin

## Abstract

This paper presents Generalized Correspondence-LDA (GC-LDA), a generalization of the Correspondence-LDA model that allows for variable spatial representations to be associated with topics, and increased flexibility in terms of the strength of the correspondence between data types induced by the model. We present three variants of GC-LDA, each of which associates topics with a different spatial representation, and apply them to a corpus of neuroimaging data. In the context of this dataset, each topic corresponds to a functional brain region, where the region's spatial extent is captured by a probability distribution over neural activity, and the region's cognitive function is captured by a probability distribution over linguistic terms. We illustrate the qualitative improvements offered by GC-LDA in terms of the types of topics extracted with alternative spatial representations, as well as the model's ability to incorporate a-priori knowledge from the neuroimaging literature. We furthermore demonstrate that the novel features of GC-LDA improve predictions for missing data.

## 1   Introduction

One primary goal of cognitive neuroscience is to find a mapping from neural activity onto cognitive processes–that is, to identify functional networks in the brain and the role they play in supporting macroscopic functions. A major milestone towards this goal would be the creation of a "functional-anatomical atlas" of human cognition, where, for each putative cognitive function, one could identify the regions and brain networks within the region that support the function.

Efforts to create such functional brain atlases are increasingly common in recent years. Most studies have proceeded by applying dimensionality reduction or source decomposition methods such as Independent Component Analysis (ICA) [4] and clustering analysis [9] to large fMRI datasets such as the Human Connectome Project [10] or the meta-analytic BrainMap database [8]. While such work has provided valuable insights, these approaches also have significant drawbacks. In particular, they typically do not jointly estimate regions along with their mapping onto cognitive processes. Instead, they first extract a set of neural regions (e.g., via ICA performed on resting-state data), and then in a separate stage—if at all—estimate a mapping onto cognitive functions. Such approaches do not allow information regarding cognitive function to constrain the spatial characterization of the regions. Moreover, many data-driven parcellation approaches involve a hard assignment of each brain voxel to a single parcel or cluster, an assumption that violates the many-to-many nature of functional brain networks. Ideally, a functional-anatomical atlas of human cognition should allow the spatial and functional correlates of each atom or unit to be *jointly* characterized, where the function of each region constrains its spatial boundaries, and vice-versa.

In the current work, we propose Generalized Correspondence LDA (GC-LDA) – a novel generalization of the Correspondence-LDA model [2] for modeling multiple data types, where one data type describes the other. While the proposed approach is general and can be applied to a variety of data, our work is motivated by its application to neuroimaging meta-analysis. To that end, we consider several GC-LDA models that we apply to the Neurosynth [12] corpus, consisting of the document text and neural activation data from a large body of neuroimaging publications. In this context, the models extract a set of neural "topics", where each topic corresponds to a functional brain region. For each topic, the model describes its spatial extent (captured via probability distributions over neural activation) and cognitive function (captured via probability distributions over linguistic terms). These models provide a novel approach for jointly identifying the spatial location and cognitive mapping of functional brain regions, that is consistent with the many-to-many nature of functional brain networks. Furthermore, to the best of our knowledge, one of the GC-LDA variants provides the first automated measure of the lateralization of cognitive functions based on large-scale imaging data.

The GC-LDA and Correspondence-LDA models are extensions of Latent Dirichlet Allocation (LDA) [3]. Several Bayesian methods with similarities (or equivalences) to LDA have been applied to different types of neuroimaging data. Poldrack et al. (2012) used standard LDA to derive topics from the text of the Neurosynth database and then projected the topics onto activation space based on document-topic loadings [7]. Yeo et al. (2014) used a variant of the Author-Topic model to model the BrainMap Database [13]. Manning et al. (2014) described a Bayesian method "Topographic Factor Analysis" to identify brain regions based on the raw fMRI images (but not text) extracted from a set of controlled experiments, which can later be mapped on functional categories [5].

Relative to the Correspondence-LDA model, the GC-LDA model incorporates: (i) the ability to associate different types of spatial distributions with each topic, (ii) flexibility in how strictly the model enforces a correspondence between the textual and spatial data within each document, and (iii) the ability to incorporate a-priori spatial structure, e.g., encouraging relatively homologous functional regions located in each brain hemisphere. As we show, these aspects of GC-LDA have a significant effect on the quality of the estimated topics, as well as on the models' ability to predict missing data.

## 2 Models

In this paper we propose a set of unsupervised generative models based on the Correspondence-LDA model [2] that we use to jointly model text and brain activations from the Neurosynth meta-analytic database [12]. Each of these models, as well as Correspondence-LDA, can be viewed as special cases of a broader model that we will refer to as Generalized Correspondence-LDA (GC-LDA). In the section below, we describe the GC-LDA model and its relationship to Correspondence-LDA. We then detail the specific instances of the model that we use throughout the remainder of the paper. A summary of the notation used throughout the paper is provided in Table 1.

### 2.1 Generalized Correspondence LDA (GC-LDA)

Each document $d$ in the corpus is comprised of two types of data: a set of word tokens $\left\{ w_1^{(d)}, w_2^{(d)}, ..., w_{N_w^{(d)}}^{(d)} \right\}$ consisting of unigrams and/or n-grams, and a set of peak activation tokens $\left\{ x_1^{(d)}, x_2^{(d)}, ..., x_{N_x^{(d)}}^{(d)} \right\}$, where $N_w^{(d)}$ and $N_x^{(d)}$ are the number of word and activation tokens in document $d$, respectively. In the target application, each token $x_i$ is a 3-dimensional vector corresponding to the peak activation coordinates of a value reported in fMRI publications. However, we note that this model can be directly applied to other types of data, such as segmented images, where each $x_i$ corresponds to a vector of real-valued features extracted from each image segment (c.f. [2]).

GC-LDA is described by the following generative process (depicted in Figure 1.A):

1. For each topic $t \in \left\{ 1, ..., T \right\}$[1]:

    (a) Sample a Multinomial distribution over word types $\phi^{(t)} \sim \text{Dirichlet}(\beta)$

2. For each document $d \in \{1, ..., D\}$:

Table 1: Table of notation used throughout the paper

| Model specification | |
|---|---|
| Notation | Meaning |
| $w_i, x_i$ | The $i$th word token and peak activation token in the corpus, respectively |
| $N_w^{(d)}, N_x^{(d)}$ | The number of word tokens and peak activation tokens in document $d$, respectively |
| $D$ | The number of documents in the corpus |
| $T$ | The number of topics in the model |
| $R$ | The number of components/subregions in each topic's spatial distribution (subregions model) |
| $z_i$ | Indicator variable assigning word token $w_i$ to a topic |
| $y_i$ | Indicator variable assigning activation token $x_i$ to a topic |
| $\mathbf{z}^{(d)}, \mathbf{y}^{(d)}$ | The set of all indicator variables for word tokens and activation tokens in document $d$ |
| $N_{td}^{YD}$ | The number of activation tokens within document $d$ that are assigned to topic $t$ |
| $c_i$ | Indicator variable assigning activation token $y_i$ to a subregion (subregion models) |
| $\Lambda^{(t)}$ | Placeholder for all spatial parameters for topic $t$ |
| $\mu^{(t)}, \sigma^{(t)}$ | Gaussian parameters for topic $t$ |
| $\mu_r^{(t)}, \sigma_r^{(t)}$ | Gaussian parameters for subregion $r$ in topic $t$ (subregion models) |
| $\phi^{(t)}$ | Multinomial distribution over word types for topic $t$ |
| $\phi_w^{(t)}$ | Probability of word type $w$ given topic $t$ |
| $\theta^{(d)}$ | Multinomial distribution over topics for document $d$ |
| $\theta_t^{(d)}$ | Probability of topic $t$ given document $d$ |
| $\pi^{(t)}$ | Multinomial distribution over subregions for topic $t$ (subregion models) |
| $\pi_r^{(t)}$ | Probability of subregion $r$ given topic $t$ (subregion models) |
| $\beta, \alpha, \gamma$ | Model hyperparameters |
| $\delta$ | Model hyperparameter (subregion models) |

(a) Sample a Multinomial distribution over topics $\theta^{(d)} \sim \text{Dirichlet}(\alpha)$

(b) For each peak activation token $x_i$, $i \in \left\{1, ..., N_x^{(d)}\right\}$:

     i. Sample indicator variable $y_i$ from Multinomial($\theta^{(d)}$)

     ii. Sample a peak activation token $x_i$ from the spatial distribution: $x_i \sim f(\Lambda^{(y_i)})$

(c) For each word token $w_i$, $i \in \left\{1, ..., N_w^{(d)}\right\}$:

     i. Sample indicator variable $z_i$ from Multinomial$\left(\frac{N_{1d}^{YD}+\gamma}{N_x^{(d)}+\gamma*T}, \frac{N_{2d}^{YD}+\gamma}{N_x^{(d)}+\gamma*T}, ...., \frac{N_{Td}^{YD}+\gamma}{N_x^{(d)}+\gamma*T}\right)$,
where $N_{td}^{YD}$ is the number of activation tokens $y$ in document $d$ that are assigned to topic $t$, $N_x^{(d)}$ is the total number of activation tokens in $d$, and $\gamma$ is a hyperparameter

     ii. Sample a word token $w_i$ from Multinomial($\phi^{(z_i)}$)

Intuitively, in the present application of GC-LDA, each topic corresponds to a functional region of the brain, where the linguistic features for the topic describe the cognitive processes associated with the spatial distribution of the topic. The resulting joint distribution of all observed peak activation tokens, word tokens, and latent parameters for each individual document in the GC-LDA model is as follows:

$$p(\mathbf{x}, \mathbf{w}, \mathbf{z}, \mathbf{y}, \theta) = p(\theta|\alpha) \cdot \left( \prod_{i=1}^{N_x^{(d)}} p(y_i|\theta^{(d)})p(x_i|\Lambda^{(y_i)}) \right) \cdot \left( \prod_{j=1}^{N_w^{(d)}} p(z_j|\mathbf{y}^{(d)}, \gamma)p(w_j|\phi^{(z_j)}) \right) \quad (1)$$

Note that when $\gamma = 0$, and the spatial distribution for each topic is specified as a single multivariate Gaussian distribution, the model becomes equivalent to a smoothed version of the Correspondence LDA model described by Blei & Jordan (2003) [2].[2]

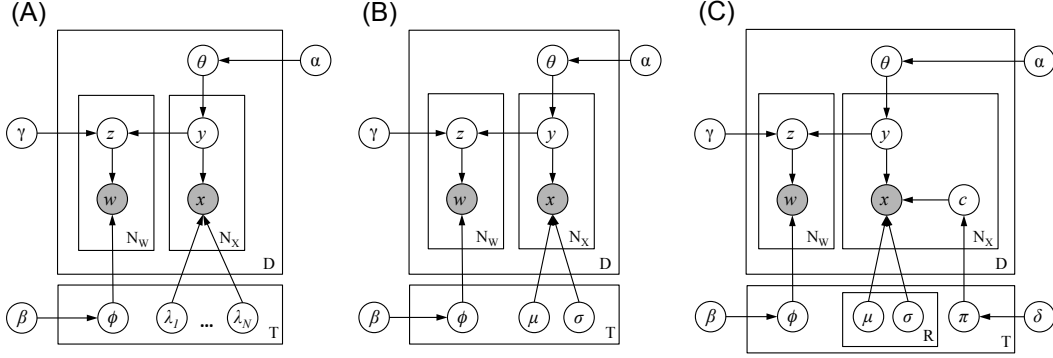

Figure 1: **(A)** Graphical model for the Generalized Correspondence-LDA model, GC-LDA. **(B)** Graphical model for GC-LDA with spatial distributions modeled as a single multivariate Gaussian (equivalent to a smoothed version of Correspondence-LDA if $\gamma = 0$)[2]. **(C)** Graphical model for GC-LDA with subregions, with spatial distributions modeled as a mixture of multivariate Gaussians

A key aspect of this model is that it induces a correspondence between the number of activation tokens and the number of word tokens within a document that will be assigned to the same topic. The hyperparameter $\gamma$ controls the strength of this correspondence. If $\gamma = 0$, then there is zero probability that a word for document $d$ will be sampled from topic $t$ if no peak activations in $d$ were sampled from $t$. As $\gamma$ becomes larger, this constraint is relaxed. Although intuitively one might want $\gamma$ to be zero in order to maximize the correspondence between the spatial and linguistic information, we have found that setting $\gamma > 0$ leads to significantly better model performance. We conjecture that using a non-zero $\gamma$ allows the parameter space to be more efficiently explored during inference, and that it improves the model's ability to handle data sparsity and noise in high dimensional spaces, similar to the role that the $\alpha$ and $\beta$ hyperparameters serve in standard LDA [1].

## 2.2 Versions of GC-LDA Employed in Current Paper

There are multiple reasonable choices for the spatial distribution $p(x_i|\Lambda^{(y_i)})$ in GC-LDA, depending upon the application and the goals of the modeler. For the purposes of the current paper, we considered three variants that are motivated by the target application. The **first model** shown in Figure 1.B employs a single multivariate Gaussian distribution for each topic's spatial distribution – and is therefore equivalent to a smoothed version of Correspondence-LDA if setting $\gamma = 0$. The generative process for this model is the same as specified above, with generative step (b.ii) modified as follows: Sample peak activation token $x_i$ from from a Gaussian distribution with parameters $\mu^{(y_i)}$ and $\sigma^{(y_i)}$. We refer to this model as the "no-subregions" model.

The **second model** and **third model** both employ Gaussian mixtures with $R = 2$ components for each topic's spatial distribution, and are shown in Figure 1.C. Employing a Gaussian mixture gives the model more flexibility in terms of the types of spatial distributions that can be associated with a topic. This is notably useful in modeling spatial distributions associated with neural activity, as it allows the model to learn topics where a single cognitive function (captured by the linguistic distribution) is associated with spatially discontiguous patterns of activations. In the second GC-LDA model we present—which we refer to as the "unconstrained subregions" model—the Gaussian mixture components are unconstrained. In the third version of GC-LDA—which we refer to as the "constrained subregions" model—the Gaussian components are constrained to have symmetric means with respect to their distance from the origin along the horizontal spatial axis (a plane corresponding to the longitudinal fissure in the brain). This constraint is consistent with results from meta-analyses of the fMRI literature, where most studied functions display a high degree of bilateral symmetry [6, 12].

The use of mixture models for representing the spatial distribution in GC-LDA requires the additional parameters $c$, $\pi$, and hyperparameter $\delta$, as well as additional modifications to the description of the generative process. Each topic's spatial distribution in these models is now associated with a multinomial probability distribution $\pi^{(t)}$ giving the probability of sampling each component $r$ from each topic $t$, where $\pi_r^{(t)}$ is the probability of sampling the $r$th component (which we will refer to as a

subregion) from the $t$th topic. Variable $c_i$ is an indicator variable that assigns each activation token $x_i$ to a subregion $r$ of the topic to which it is assigned via $y_i$. A full description of the generative process for these models is provided in Section 1 of the supplementary materials[3].

## 2.3 Inference for GC-LDA

Exact probabilistic inference for the GC-LDA model is intractable. We employed collapsed Gibbs sampling for posterior inference – collapsing out $\theta^{(d)}$, $\phi^{(t)}$, and $\pi^{(t)}$ while sampling the indicator variables $y_i$, $z_i$ and $c_i$. Spatial distribution parameters $\Lambda^{(t)}$ are estimated via maximum likelihood. The per-iteration computational complexity of inference is $O(T(N_W + N_X R))$, where $T$ is the number of topics, $R$ is the number of subregions, and $N_W$ and $N_X$ are the total number of word tokens and activation tokens in the corpus, respectively. Details of the inference methods and sampling equations are provided in Section 2 of the supplement.

# 3 Experimental Evaluation

We refer to the three versions of GC-LDA described in Section 2 as (1) the "no subregions" model, for the model in which each topic's spatial distribution is a single multivariate Gaussian distribution, (2) the "unconstrained subregions" model, for the model in which each topic's spatial distribution is a mixture of $R = 2$ unconstrained Gaussian distributions, and (3) the "constrained subregions" model, for the model in which each topic's spatial distribution is a mixture of $R = 2$ Gaussian distributions whose means are constrained to be symmetric along the horizontal spatial dimension with respect to their distance from the origin.

Our empirical evaluations of the GC-LDA model are based on the application of these models to the Neurosynth meta-analytic database [12]. We first illustrate and contrast the qualitative properties of topics that are extracted by the three versions of GC-LDA[4]. We then provide a quantitative model comparison, in which the models are evaluated in terms of their ability to predict held out data. These results highlight the promise of GC-LDA and this type of modeling for jointly extracting the spatial extent and cognitive functions of neuroanatomical brain regions.

**Neurosynth Database:** Neurosynth [12] is a publicly available database consisting of data automatically extracted from a large collection of functional magnetic resonance imaging (fMRI) publications[5]. For each publication, the database contains the abstract text and all reported 3-dimensional peak activation coordinates (in MNI space) in the study. The text was pre-processed to remove common stop-words. For the version of the Neurosynth database employed in the current paper, there were 11,362 total publications, which had on average 35 peak activation tokens and 46 word tokens after preprocessing (corresponding to approximately 400k activation and 520k word tokens in total).

## 3.1 Visualizing GC-LDA Topics

In Figure 2 we present several illustrative examples of topics for all three GC-LDA variants that we considered. For each topic, we illustrate the topic's distribution over word types via a word cloud, where the sizes of words are proportional to their probabilities $\phi_w^{(t)}$ in the model. Each topic's spatial distribution over neural activations is illustrated via a kernel-smoothed representation of all activation tokens that were assigned to the topic, overlaid on an image of the brain. For the models that represent spatial distributions using Gaussian mixtures (the unconstrained and constrained subregions models), activations are color-coded based on which subregion they are assigned to, and the mixture weights for the subregions $\pi_r^{(t)}$ are depicted above the activation image on the left. In the constrained subregions model (where the means of the two Gaussians were constrained to be symmetric along the horizontal axis) the two subregions correspond to a 'left' and 'right' hemisphere subregion. The following parameter settings were used for generating the images in Figure 2: $T = 200$, $\alpha = .1$, $\beta = .01$, $\gamma = .01$, and for the models with subregions, $\delta = 1.0$.

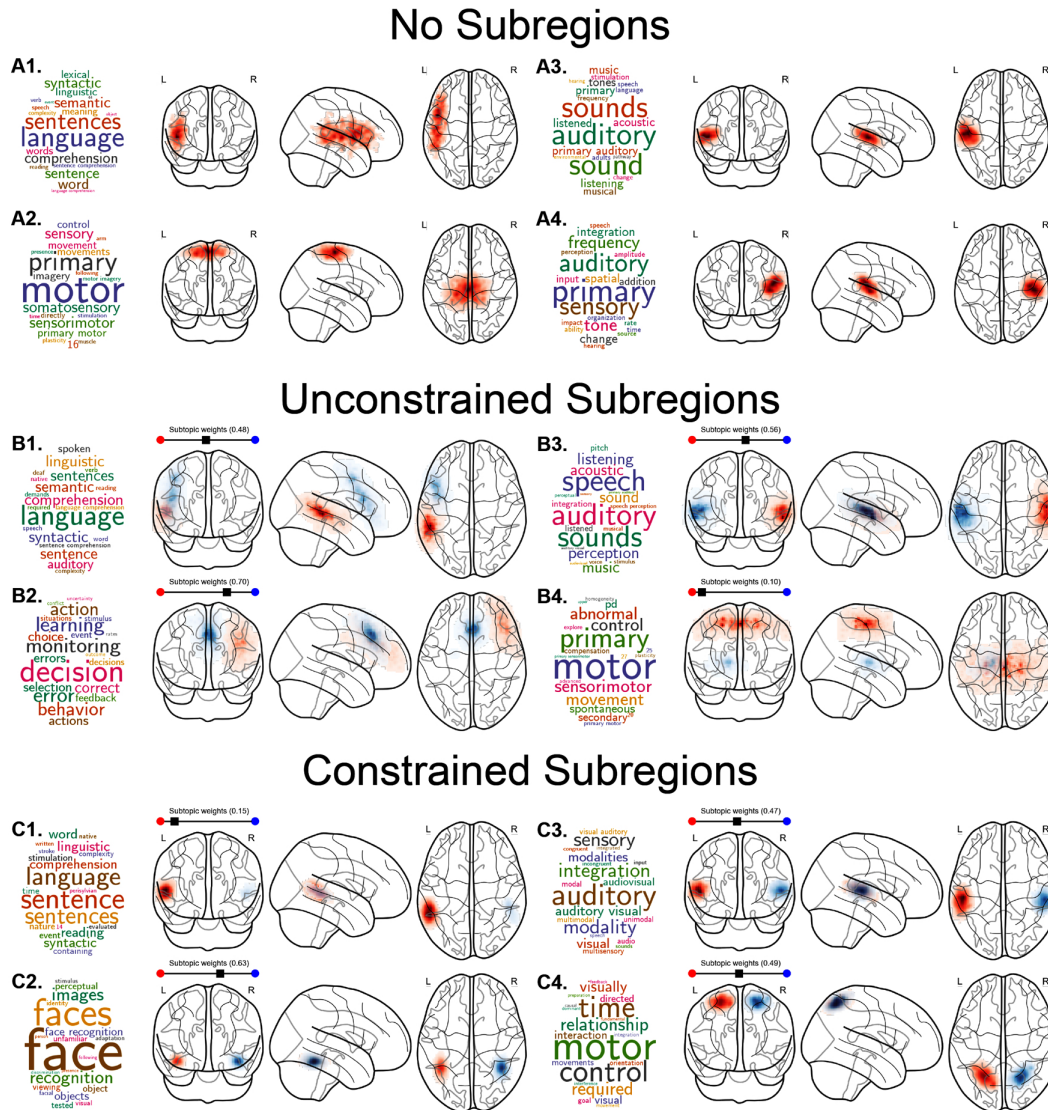

Figure 2: Illustrative examples of topics extracted for the three GC-LDA variants. Probability distributions over word types $\phi^{(t)}$ are represented via word clouds, where word sizes are proportional to $\phi_w^{(t)}$. Spatial distributions are illustrated using kernel-smoothed representations of all activation tokens assigned to each topic. For the models with subregions, each activation token's color (blue or red) corresponds to the subregion $r$ that the token is assigned to.

For nearly all of the topics shown in Figure 2, the spatial and linguistic distributions closely correspond to functional regions that are extensively described in the literature (e.g., motor function in primary motor cortex; face processing in the fusiform gyrus, etc.). We note that a key feature of all versions of the GC-LDA model, relative to the majority of existing methods in the literature, is that the model is able to capture the one-to-many mapping from neural regions onto cognitive functions. For example, in all model variants, we observe topics corresponding to auditory processing and language processing (e.g., the topics shown in panels B1 and B3 for the subregions model). While these cognitive processes are distinct, they have partial overlap with respect to the brain networks they recruit – specifically, the superior temporal sulcus in the left hemisphere.

For functional regions that are relatively medial, the no-subregions model is able to capture bilateral homologues by consolidating them into a single distribution (e.g., the topic shown in A2, which spans the medial primary somatomotor cortex in both hemispheres). However, for functional regions that are more laterally localized, the model cannot capture bilateral homologues using a single topic. For cognitive processes that are highly lateralized (such as language processing, shown in A1, B1

and C1) this poses no concern. However, for functional regions that are laterally distant and do have spatial symmetry, the model ends up distributing the functional region across multiple topics–see, e.g., the topics shown in A3 and A4 in the no-subregions model, which correspond to the auditory cortex in the left and right hemisphere respectively. Given that these two topics (and many other pairs of topics that are not shown) correspond to a single cognitive function, it would be preferable if they were represented using a single topic. This can potentially be achieved by increasing the flexibility of the spatial representations associated with each topic, such that the model can capture functional regions with distant lateral symmetry or other discontiguous spatial features using a single topic. This motivates the unconstrained and constrained subregions models, in which topic's spatial distributions are represented by Gaussian mixtures.

In Figure 2, the topics in panels B3 and C3 illustrate how the subregions models are able to handle symmetric functional regions that are located on the lateral surface of the brain. The lexical distribution for each of these individual topics in the subregions models is similar to that of both the topics shown in A3 and A4 of the no-subregions model. However, the spatial distributions in B3 and C3 each capture a summation of the two topics from the no subregions model. In the case of the constrained subregion model, the symmetry between the means of the spatial distributions for the subregions is enforced, while for the unconstrained model the symmetry is data-driven and falls out of the model.

We note that while the unconstrained subregions model picks up spatial symmetry in a significant subset of topics, it does not always do so. In the case of language processing (panel A1), the lack of spatial symmetry is consistent with a large fMRI literature demonstrating that language processing is highly left-lateralized [11]. And in fact, the two subregions in this topic correspond approximately to Wernicke's and Broca's areas, which are integral to language comprehension and production, respectively. In other cases, (e.g., the topics in panels B2 and B4), the unconstrained subregions model partially captures spatial symmetry with a highly-weighted subregion near the horizontal midpoint, but also has an additional low-weighted region that is lateralized. While this result is not necessarily wrong per se, it is somewhat inelegant from a neurobiological standpoint. Moreover, there are theoretical reasons to prefer a model in which subregions are always laterally-symmetrical. Specifically, in instances where the subregions are symmetric (the topic in panel B3 for the unconstrained subregions model and all topics for the constrained subregions model), the subregion weights provide a measure of the relative lateralization of function. For example, the language topic in panel C1 of the constrained subregions model illustrates that while there is neural activation corresponding to linguistic processing in the right hemisphere of the brain, the function is strongly left-lateralized (and vice-versa for face processing, illustrated in panel C2). By enforcing the lateral symmetry in the constrained subregions model, the subregion weights $\pi_r^{(t)}$ (illustrated above the left activation images) for each topic inherently correspond to an automated measure of the lateralization of the topic's function. Thus, the constrained model produces what is, to our knowledge, the first data-driven estimation of region-level functional hemispheric asymmetry across the whole brain.

### 3.2 Predicting Held Out Data

This section describes quantitative comparisons between three GC-LDA models in terms of their ability to predict held-out data. We split the Neurosynth dataset into a training and test set, where approximately 20% of all data in the corpus was put into the test set. For each document, we randomly removed $\left\lfloor .2N_x^{(d)} \right\rfloor$ peak activation tokens and $\left\lfloor .2N_w^{(d)} \right\rfloor$ word tokens from each document. We trained the models on the remaining data, and then for each model we computed the log-likelihood of the test data, both for the word tokens and peak tokens.

The space of possible hyperparameters to explore in GC-LDA is vast, so we restrict our comparison to the aspects of the model which are novel relative to the original Correspondence-LDA model. Specifically, for all three model variants, we compared the log-likelihood of the test data across different values of $\gamma$, where $\gamma \in \{0, 0.001, 0.01, 0.1, 1\}$. We note again here that the no-subregions model with $\gamma = 0$ is equivalent to a smoothed version of Correspondence-LDA [2] (see footnote 2 for additional clarification). The remainder of the parameters were fixed as follows (chosen based on a combination of precedent from the topic modeling literature and preliminary model exploration): $T = 100$, $\alpha = .1$, and $\beta = .01$ for all models, and $\delta = 1.0$ for the models with subregions. All models were trained for 1000 iterations.

Figure 3 presents the held out log-likelihoods for all models across different settings of $\gamma$, in terms of (i) the total log-likelihood for both activation tokens and word tokens (left) (ii) log-likelihood for activation tokens only (middle), and (iii) log likelihood for word tokens only (right). For both activation tokens and word tokens, for all three versions of GC-LDA, using a non-zero $\gamma$ leads to significant improvement in performance. In terms of predicting activation tokens alone, there is a monotonic relationship between the size of $\gamma$ and log-likelihood. This is unsurprising, since increasing $\gamma$ reduces the extent that word tokens constrain the spatial fit of the model. In terms of predicting word tokens (and overall log-likelihood), the effect of $\gamma$ shows an inverted-U function, with the best performance in the range of .01 to .1. These patterns were consistent across all three variants of GC-LDA. Taken together, our results suggest that using a non-zero $\gamma$ results in a significant improvement over the Correspondence-LDA model.

In terms of comparisons across model variants, we found that both subregions models were significant improvements over the no-subregions models in terms of total log-likelihood, although the no-subregions model performed slightly better than the constrained subregions model at predicting word tokens. In terms of the two subregions models, performance is overall fairly similar. Generally, the constrained subregions model performs slightly better than the unconstrained model in terms of predicting peak tokens, but slightly worse in terms of predicting word tokens. The differences between the two subregions models in terms of total log-likelihood were negligible. These results do not provide a strong statistical case for choosing one subregions model over the other; instead, they suggest that the modeler ought to choose between models based on their respective theoretical or qualitative properties (e.g., biological plausibility, as discussed in Section 3.1).

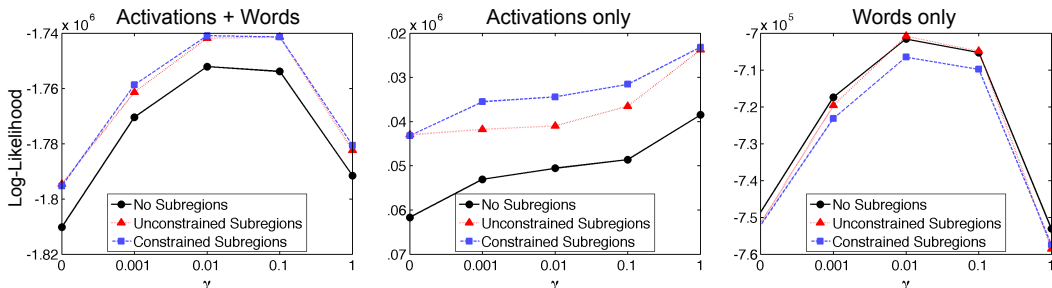

Figure 3: Log Likelihoods of held out data for the three GC-LDA models as a function of model parameter $\gamma$. Left: total log-likelihood (activation tokens + word tokens). Middle: log-likelihood of activation tokens only. Right: log-likelihood of word tokens only.

## 4   Summary

We have presented generalized correspondence LDA (GC-LDA) – a generalization of the Correspondence-LDA model, with a focus on three variants that capture spatial properties motivated by neuroimaging applications. We illustrated how this model can be applied to a novel type of metadata—namely, the spatial peak activation coordinates reported in fMRI publications—and how it can be used to generate a relatively comprehensive atlas of functional brain regions. Our quantitative comparisons demonstrate that the GC-LDA model outperforms the original Correspondence-LDA model at predicting both missing word tokens and missing activation peak tokens. This improvement was demonstrated in terms of both the introduction of the $\gamma$ parameter, and with respect to alternative parameterizations of topics' spatial distributions.

Beyond these quantitative results, our qualitative analysis demonstrates that the model can recover interpretable topics corresponding closely to known functional regions of the brain. We also showed that one variant of the model can recover known features regarding the hemispheric lateralization of certain cognitive functions. These models show promise for the field of cognitive neuroscience, both for summarizing existing results and for generating novel hypotheses. We also expect that novel features of GC-LDA can be carried over to other extensions of Correspondence-LDA in the literature.

In future work, we plan to explore other spatial variants of these models that may better capture the morphological features of distinct brain regions – e.g., using hierarchical priors that can capture the hierarchical organization of brain systems. We also hope to improve the model by incorporating features such as the correlation between topics. Applications and extensions of our approach for more standard image processing applications may also be a fruitful area of research.

## Footnotes

[1]To make the model fully generative, one could additionally put a prior on the spatial distribution parameters $\Lambda^{(t)}$ and sample them. For the purposes of the present paper we do not specify a prior on these parameters, and therefore leave this out of the generative process.

[2]We note that [2] uses a different generative description for how the $z_i$ variables are sampled conditional on the $y_i^{(d)}$ indicator variables; in [2], $z_i$ is sampled uniformly from $(1, ..., N_y^{(d)})$, and then $w_i$ is sampled from the multinomial distribution of the topic $y_i^{(d)}$ that $z_i$ points to. This ends up being functionally equivalent to the generative description for $z_i$ given here when $\gamma = 0$. Additionally, in [2], no prior is put on $\phi^{(t)}$, unlike in GC-LDA. Therefore, when using GC-LDA with a single multivariate Gaussian and $\gamma = 0$, it is equivalent to a *smoothed* version of Correspondence-LDA. Dirichlet priors have been demonstrated to be beneficial to model performance [1], so including a prior on $\phi^{(t)}$ in GC-LDA should have a positive impact.

[3]Note that these models are still instances of GC-LDA as presented in Figure 1.1; they can be equivalently formulated by marginalizing out the $c_i$ variables, such that the probability $f(x_i|\Lambda^{(t)})$ depends directly on the parameters of each component, and the component probabilities given by $\pi^{(t)}$.

[4]A brief discussion of the stability of topics extracted by GC-LDA is provided in Section 3 of the supplement

[5]Additional details and Neurosynth data can be found at `http://neurosynth.org/`

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
