[Supplementary Material · NIPS_2016_GCLDA_Supplement_CameraReady.pdf]

# Supplement: Generalized Correspondence-LDA Models (GC-LDA) for Identifying Functional Regions in the Brain

Section 1 of this supplement presents the generative process for the GC-LDA variants that use Gaussian mixtures to model each topic's spatial component. Section 2 provides inference details for all versions of GC-LDA considered in this paper[1]. Section 3 provides an analysis of the stability of the topics extracted by GC-LDA.

The notation used for both model specification and inference throughout the supplement is summarized in Table 1.

## 1 Generative Process and Joint Distribution for GC-LDA with Gaussian Mixtures

For completeness, we present here a modified version of the generative process for the GC-LDA models in which the spatial distributions are modeled as mixtures of multivariate Gaussians with $R$ components. We only present the updated process for generating topics $t$ and activation tokens $x_i$, as the generative process for sampling word tokens $w_i$ does not depend on the parameterization of the spatial distributions:

1. For each topic $t \in \{1, ..., T\}$:

    (a) Sample a Multinomial distribution over word types $\phi^{(t)} \sim$ Dirichlet($\beta$)

    (b) Sample a Multinomial distribution over subregions $\pi^{(t)} \sim$ Dirichlet($\delta$)

2. For each document $d \in \{1, ..., D\}$:

    (a) For each peak activation token $x_i \in \{1, ..., N_x^{(d)}\}$:

        i. Sample indicator variable $y_i$ from Multinomial($\theta^{(d)}$)
        ii. Sample indicator variable $c_i$ from Multinomial($\pi^{(y_i)}$)
        iii. Sample a peak activation token $x_i$ from the spatial distribution for subregion $r_{c_i}^{(y_i)}$: $x_i \sim$ Gaussian($\mu_{c_i}^{(y_i)}, \sigma_{c_i}^{(y_i)}$)

The joint distribution of all observed peak activation tokens, word tokens, and latent parameters for each individual document in the GC-LDA model with a mixture of Gaussian spatial distributions is as follows:

$$p(\mathbf{x}, \mathbf{w}, \mathbf{z}, \mathbf{y}, \mathbf{c}, \theta) = p(\theta|\alpha) \cdot \left( \prod_{i=1}^{N_x^{(d)}} p(y_i|\theta^{(d)}) p(c_i|\pi^{(y_i)}) p(x_i|\mu_{c_i}^{(y_i)}, \sigma_{c_i}^{(y_i)}) \right) \cdot \left( \prod_{j=1}^{N_w^{(d)}} p(z_j|\mathbf{y}^{(d)}, \gamma) p(w_j|\phi^{(z_j)}) \right)$$

$$(1)$$

Figure 1: **(A)** Plate notation for the Generalized Correspondence-LDA model, GC-LDA. **(B)** Plate notation for GC-LDA with spatial distributions modeled as a single multivariate Gaussian (Equivalent to a smoothed version of Correspondence-LDA if $\gamma = 0$). **(C)** Plate notation for GC-LDA with subregions, with spatial distributions modeled as a mixture of multivariate Gaussians

## 2 Inference for GC-LDA

During inference, we seek to estimate the posterior distribution across all unobserved model parameters. As is typical with topic models, exact probabilistic inference for the GC-LDA model is intractable. Inference for the original Correspondence LDA model [2] used Variational Bayesian methods. Here, we employ a mixture of MCMC techniques based on Gibbs Sampling [4], since Gibbs sampling approaches have often outperformed variational methods for inference in LDA [1, 5]. The per-iteration computational complexity is $O(T(N_W + N_X R))$, where $T$ is the number of topics, $R$ the number of subregions, and $N_W$ and $N_X$ are the total number of word tokens and activation tokens in the corpus, respectively.

In describing the inference procedure, we will provide update equations for the three variants of GC-LDA that were used in our experiments, depicted in Figures 1.B and 1.C. Specifically, we describe the updates for the GC-LDA model where each topic's spatial component is represented by a single Gaussian distribution (Figure 1.B), and for the two GC-LDA models where each topic's spatial distribution is represented by a mixture of Gaussian distributions (Figure 1.C). As a reminder, the difference between the two versions of the model that use Gaussian mixtures (referred to as the "unconstrained subregions" and "constrained subregions" models), is that we constrain the mean of the two Gaussian components to be symmetric with respect to their distance from the origin along the horizontal spatial axis in the "constrained subregions" model. In places where the updates for the versions of the models are different, we will first describe the update for the model with single a Gaussian distribution, and then describe how it is modified for the models that use Gaussian mixtures.

After model initialization, our Gibbs Sampling method involves sequentially updating the spatial distribution parameters $\Lambda^{(t)}$ for all topics, the assignments $z_i$ of word tokens to topics, and the assignments $y_i$ of peak activation tokens to topics (and additionally the assignments $c_i$ of activation tokens to subregions when using a Gaussian mixture model for each topic's spatial distribution). We first provide an overview of the sampling algorithm sequence, and then describe in detail the update equations used at each step. We also note here that the update equations presented here will generalize to any variant of the GC-LDA model using a single parametric or mixture of parametric spatial distributions, provided the updates for the spatial parameter estimates are modified appropriately.

### 2.1 Overview of Inference Procedure

Configuring and running the model consists of two phases: (1) Model initialization, and (2) Inference. We first describe model initialization, and give an overview of the sequence in which model parameters are updated. We will then provide the exact update equations for each of the steps used during inference.

Table 1: Table of notation used throughout the appendix

| Model specification | |
|---|---|
| **Notation** | **Meaning** |
| $w_i, x_i$ | The $i$th word token and peak activation token in the corpus |
| $N_x^{(d)}, N_w^{(d)}$ | The number of word tokens and peak activation tokens in document $d$, respectively |
| $D$ | The number of documents in the corpus |
| $T$ | The number of topics in the model |
| $R$ | The number of components/subregions in each topic's spatial distribution (subregions model) |
| $z_i$ | Indicator variable assigning word token $w_i$ to a topic |
| $y_i$ | Indicator variable assigning activation token $x_i$ to a topic |
| $\mathbf{z}^{(d)}, \mathbf{y}^{(d)}$ | The set of all indicator variables for word tokens and activation tokens in document $d$ |
| $N_{td}^{YD}$ | The number of activation tokens within document $d$ that are assigned to topic $t$ |
| $c_i$ | Indicator variable assigning activation token $y_i$ to a subregion (subregion models) |
| $\Lambda^{(t)}$ | Placeholder for all spatial parameters for topic $t$ |
| $\mu^{(t)}, \sigma^{(t)}$ | Gaussian parameters for topic $t$ |
| $\mu_r^{(t)}, \sigma_r^{(t)}$ | Gaussian parameters for subregion $r$ in topic $t$ (subregion models) |
| $\phi^{(t)}$ | Multinomial distribution over word types for topic $t$ |
| $\phi_w^{(t)}$ | Probability of word type $w$ given topic $t$ |
| $\theta^{(d)}$ | Multinomial distribution over topics for document $d$ |
| $\theta_t^{(d)}$ | Probability of topic $t$ given document $d$ |
| $\pi^{(t)}$ | Multinomial distribution over subregions for topic $t$ (subregion models) |
| $\pi_r^{(t)}$ | Probability of subregion $r$ given topic $t$ (subregion models) |
| $\beta, \alpha, \gamma$ | Model hyperparameters |
| $\delta$ | Model hyperparameter (subregion models) |
| **Count matrices used during model inference** | |
| **Notation** | **Meaning** |
| $N_{t\cdot}^{YT}$ | The number of activation tokens that are assigned via $y_i$ to topic $t$ |
| $N_{td,-i}^{YD}$ | The number of activation tokens in document $d$ that are assigned via $y_i$ to topic $t$, excluding the $i$th token |
| $N_{z_j d}^{YD*}$ | The number of activation tokens in document $d$ that would be assigned to the topic indicated by $z_j$, given the proposed update of $y_i$ |
| $N_{rt,-i}^{CT}$ | The number of activation tokens that are assigned via $c_i$ to subregion $r$ in topic $t$, excluding the $i$th token (subregion models) |
| $N_{wt,-i}^{ZT}$ | The number of times word type $w$ is assigned via $z_i$ to topic $t$, excluding the $i$th token |
| $N_{td}^{ZD}$ | The number of word tokens in document $d$ that are assigned via $z_i$ to topic $t$ |

### 2.1.1 Model Initialization

To initialize the model, we first randomly assign all $y_i$ indicator variables to one of the topics $y_i \sim \text{uniform}(1, ..., T)$. The $z_i$ indicator variables are randomly sampled from the multinomial distribution conditioned on $y_i^{(d)}$ as defined in the generative model: $z_i \sim \text{Multinomial}\left( \frac{N_{1d}^{YD}+\gamma}{N_x^{(d)}+\gamma*T}, \frac{N_{2d}^{YD}+\gamma}{N_x^{(d)}+\gamma*T}, ..., \frac{N_{Td}^{YD}+\gamma}{N_x^{(d)}+\gamma*T} \right)$. In the model that uses an unconstrained mixture of Gaussians with $R = 2$, the initial $c_i$ are randomly assigned: $c_i \sim \text{uniform}(1, ...R)$. In "constrained subregions" model we used a deterministic initial assignment, where we set $c_i = 1$ if the x-coordinate of the activation token was less than or equal to zero (i.e., if the activation peak fell within the left hemisphere of the brain), and $c = 2$ otherwise.

### 2.1.2 Parameter Update Sequence

After initialization, the model inference procedure entails repeating the following three parameter update steps until the algorithm has converged:

1. For each topic $t$, update the estimate of the spatial distribution parameters $\Lambda^{(t)}$ conditioned on the subset of peaks $x_i$ with indicator variables $y_i = t$. When using a model with subregions for the topics' spatial components, update the estimate of the spatial distribution

parameters $\Lambda_r^{(t)}$ conditioned on the subsets of peaks $x_i$ with indicator variables $y_i = t$ and $c_i = r$.

2. For each activation token $x_i$ in each document $d$, update the corresponding indicator variable $y_i$ assigning the token to a topic, conditioned on the current estimates of all spatial distribution parameters $\Lambda^{(\cdot)}$, the current assignments of $\mathbf{z}^{(d)}$ of all word tokens to topics in document $d$, and the current estimate of the document's multinomial distribution over topics $\theta^{(d)}$. When using a model with subregions, instead jointly update the indicator variables $y_i$ of the token to a topic and $c_i$ of the token to a subregion within topic $y_i$. This update is additionally conditioned on the current estimate of all topic's multinomial distributions over subregions $\pi^{(\cdot)}$.

3. For each word token $w_i$ in each document $d$, update the corresponding indicator variable $z_i$ assigning the token to a topic, conditioned on the current estimates of all topics' multinomial distributions over words $\phi^{(\cdot)}$, and the current assignments $\mathbf{y}^{(d)}$ of all peaks to topics in document $d$.

Note that we do not need to directly update the $\theta^{(d)}$, $\phi^{(t)}$ or $\pi^{(t)}$ parameters during inference, because these distributions are "collapsed out" [5] and are estimated directly from the current state of indicator variables $\mathbf{y}$, $\mathbf{z}$, and $\mathbf{c}$, respectively. Convergence of this algorithm is evaluated by computing the log-likelihood of the observed data after every iteration of the sampler; when the log-likelihood is no increasing over multiple iterations, we halt the algorithm and compute a final estimate of all parameters.

We now provide the update equations for each of these steps.

## 2.2 Updating Spatial Distribution Estimates: $\Lambda^{(t)}$

To estimate the spatial distributions, we compute the maximum likelihood estimates of the spatial distribution for each topic $t$, conditioned on the subset of peak activation tokens that are assigned to $t$. When each topic is associated with a single multivariate Gaussian distribution:

$$\hat{\mu}^{(t)} = \frac{\sum_{i, y_i = t} x_i}{N_{t\cdot}^{YT}} \tag{2}$$

$$\hat{\sigma}^{(t)} = \frac{\sum_{i, y_i = t} (x_i - \hat{\mu}^{(t)})^2}{N_{t\cdot}^{YT}} \tag{3}$$

where $N_{t\cdot}^{YT}$ is the total number of peak activation tokens $x_i$ that are assigned (via $y_i$) to $t$. When using a mixture of Gaussians for the spatial distributions, the same estimates are used to estimate the means and covariances for each subregion, $\hat{\mu}_r^{(t)}$ and $\hat{\sigma}_r^{(t)}$, except that the sums are computed over the subset of peak activation tokens for which $y_i = t$ and $c_i = r$. Similarly, for any arbitrary choice of spatial distribution not specifically considered in this paper (e.g., a kernel density estimator), one can use the standard maximum likelihood estimator.

In the "constrained subregions" model, where the Gaussian component means are constrained to be symmetric about the horizontal spatial axis (with respect to the distance from the origin), we must further modify the estimation procedure. We estimate a single mean for the two subregions, with respect to it's location along the horizontal axis in terms of distance from the origin (corresponding to the longitudinal fissure of the brain), by computing the average coordinates of all $x_i$ tokens that are assigned to $t$ after taking the absolute value of the tokens' distance from the origin. This estimate is then used as the mean of the 2nd subregion along the horizontal axis, and the mean of the 1st subregion is set equal to the same mean, reflected about the horizontal axis (so that along this coordinate, $\hat{\mu}_1^{(t)} = -\hat{\mu}_2^{(t)}$). The covariance matrices of the two subregions are estimated independently using equation 3. We note that these updates correspond to maximum likelihood estimates, subject to the constraint that the mean is symmetric along the horizontal axis.

## 2.3 Updating Assignments $y_i$ of Activation Tokens $x_i$ to Topics

This update step, in which peak activation tokens $x_i$ to are assigned to topics via the indicator variables $y_i$, is dependent upon the choice of the spatial distribution. Specifically, when using a

model with topic subregions (e.g., where each topic is associated with a Gaussian mixture), this step involves additionally updating the $c_i$ assignments of tokens to subregions. We first provide the update equations for the model that uses a Gaussian distribution for each topic, and then describe the modification to this update needed when using a subregions model.

### 2.3.1 Updating $y_i$ Assignments for GC-LDA Models Using Single Multivariate Gaussian Spatial Distributions

Here, we wish to update the indicator variable $y_i^{(d)}$, which is the assignment of the $i$th peak activation token $x_i$ of document $d$ to a topic. This update is conditioned on the current estimates of all spatial distribution parameters $\Lambda$, the current vector $\mathbf{z}^{(d)}$ of assignments of words to topics in document $d$, and the current estimate of the document's multinomial distribution over topics $\theta^{(d)}$.

We employ a Gibbs Sampling step to update each indicator variable using a proposal distribution. The proposal distribution is used to compute the relative probabilities that $x_i$ should be assigned to a specific topic $t = 1, ...T$. Once the relative probabilities are computed across all topics, we randomly sample a topic-assignment $y_i$ from the proposal distribution, normalized such that the probability of assigning the word to a topic sums to 1 across all topics. The update equation is as follows:

$$p(y_i = t | x_i, \mathbf{z}^{(d)}, \mathbf{y}_{-i}^{(d)}, \Lambda^{(t)}, \gamma, \alpha) \sim p(x_i | \Lambda^{(t)}) \cdot p(t | \theta^{(d)}) \cdot p(\mathbf{z}^{(d)} | \mathbf{y}^{(d)*}, \gamma)$$

$$\sim p(x_i | \Lambda^{(t)}) \cdot (N_{td,-i}^{YD} + \alpha) \cdot \prod_{j=1}^{N_w^{(d)}} \frac{N_{z_j d}^{YD*} + \gamma}{N_x^{(d)} + \gamma * T} \qquad (4)$$

$$\sim p(x_i | \Lambda^{(t)}) \cdot (N_{td,-i}^{YD} + \alpha) \cdot \left( \frac{N_{td,-i}^{YD} + \gamma + 1}{N_{td,-i}^{YD} + \gamma} \right)^{N_{td}^{ZD}}$$

To understand this equation and the notation, we consider the three main terms in the equation in detail.

The first term, $p(x_i | \Lambda^{(t)})$, is the probability that peak activation $x_i$ was generated from the spatial distribution associated with topic $t$. For example, if each topic is associated with a single multivariate Gaussian distribution, this term corresponds to the multivariate Gaussian probability density function with parameters $\mu^{(t)}$ and $\sigma^{(t)}$ evaluated at location $x_i$.

The second term, $(N_{td,-i}^{YD} + \alpha)$ is an estimate of the probability of sampling topic $t$ from $\theta^{(d)}$, using an estimate of $\theta^{(d)}$ that is computed from the set of all indicator variables $\mathbf{y}_{-i}^{(d)}$ in document $d$ excluding the indicator variable for the token $i$ that is currently being sampled. In the notation above, $N_{td,-i}^{YD}$ is equal to the number of activation tokens in document $d$ that are currently assigned via $y$ to topic $t$, where $-i$ indicates that the current token that we are sampling is removed from these counts.

The third term, $\prod_{j=1}^{N_w^{(d)}} \frac{N_{z_j d}^{YD*} + \gamma}{N_x^{(d)} + \gamma * T}$ is the multinomial probability of sampling all of the current indicator variables $\mathbf{z}^{(d)}$ for words in document $d$, given the count matrix $N_{\cdot d}^{YD*}$ that results from the proposed update of the indicator variables for the peak assignment $y_i$. In this notation, $\frac{N_{z_j d}^{YD*} + \gamma}{N_x^{(d)} + \gamma * T}$ is the multinomial probability of sampling the indicator variable $z_j$ from the proposed vector of peak-topic assignments $\mathbf{y}^{(d)*}$, where $N_{z_j d}^{YD*}$ is the number of $y$ indicator variables that would be assigned to the same topic as indicator variable $z_j$ given the proposed update of $y_i$. In the context of Gibbs sampling, the third term can be simplified as shown in the final form of the equation, in which $N_{td}^{ZD}$ corresponds to the number of word tokens in document $d$ that are currently assigned via $z$ to topic $t$.

### 2.3.2 Updating $y_i$ and $c_i$ Assignments for GC-LDA models Using Mixtures of Multivariate Gaussian Spatial Distributions

In the GC-LDA model in which each topic's spatial distribution is a mixture of multivariate Gaussian distributions, we use a modified Gibbs sampling procedure in which we jointly sample both the $y_i$ assignment of the peak activation token to a topic, and the $c_i$ assignment of the peak activation token to a subregion, according to the following update update equation:

$$p(y_i = t, c_i = r | x_i, \mathbf{z}^{(d)}, \mathbf{y}_{-i}^{(d)}, \Lambda_r^{(t)}, \pi^{(t)}, \delta, \gamma, \alpha)$$

$$\sim p(x_i | \Lambda_r^{(t)}) \cdot p(t | \theta^{(d)}) \cdot p(r | \pi^{(t)}) \cdot p(\mathbf{z}^{(d)} | \mathbf{y}^{(d)*}, \gamma)$$

$$\sim p(x_i | \Lambda_r^{(t)}) \cdot (N_{td,-i}^{YD} + \alpha) \cdot \frac{N_{rt,-i}^{CT} + \delta}{\sum_{r'=1}^{R}(N_{r't,-i}^{CT} + \delta)} \cdot \prod_{j=1}^{N_w^{(d)}} \frac{N_{z_j d}^{YD*} + \gamma}{N_x^{(d)} + \gamma * T} \tag{5}$$

$$\sim p(x_i | \Lambda_r^{(t)}) \cdot (N_{td,-i}^{YD} + \alpha) \cdot \frac{N_{rt,-i}^{CT} + \delta}{\sum_{r'=1}^{R}(N_{r't,-i}^{CT} + \delta)} \cdot \left( \frac{N_{td,-i}^{YD} + \gamma + 1}{N_{td,-i}^{YD} + \gamma} \right)^{N_{td}^{ZD}}$$

This update equation is the same as the update equation for the model with a single multivariate Gaussian distribution per topic, with the exception of the first and third terms. The first term $p(x_i | \Lambda_r^{(t)})$ now corresponds the probability that peak activation $x_i$ was generated from the spatial distribution associated with subregion $r$ of topic $t$. The third term is the probability $\pi_r^{(t)}$ of sampling subregion $r$ from topic $t$. The notation $N_{rt,-i}^{CT}$ corresponds to the total number of subregion indicator variables $c_i$ that are currently assigned to subregion $r$ within topic $t$, excluding the count of the token that is currently being sampled.

## 2.4 Updating $z_i$ Assignments of Word Tokens $w_i$ to topics

Here we wish to update the indicator variables $z_i^{(d)}$, giving the assignment of the $i$th word token $w_i$ in document $d$ to a topic. This update is conditioned on the current vector $\mathbf{y}^{(d)}$ of assignments of peaks to topics in $d$, and an estimate of each topic's multinomial distribution over word types $\phi^{(t)}$

This update involves a collapsed Gibbs sampling step similar in form to the one employed for inference in standard LDA [5]. The update equation is as follows:

$$p(z_i = t | w_i, \mathbf{z}_{-i}, \mathbf{y}^{(d)}, \gamma, \beta) \sim p(t | \mathbf{y}^{(d)}, \gamma) \cdot p(w_i | \phi^{(t)})$$

$$\sim (N_{td,-i}^{YD} + \gamma) \cdot \frac{N_{wt,-i}^{ZT} + \beta}{\sum_{w'=1}^{T}(N_{w't,-i}^{ZT} + \beta)} \tag{6}$$

The first term in this equation gives the probability of sampling topic $t$ from document $d$, which is proportional to $N_{td,-i}^{YD}$—the count of the number of activation tokens in document $d$ that are currently assigned to topic $t$—plus the smoothing parameter $\gamma$, as defined in the generative model. The second term in this equation is the probability of sampling word $w_i$ from topic $t$, given the current estimates of the topic-word multinomial distributions. As with the estimate of $\theta^{(d)}$ computed during the $\mathbf{y}_i$ update steps, $\phi^{(t)}$ is computed from the counts of word token assignments, where $N_{wt,-i}^{ZT}$ is the number of times word type $w$ is assigned to topic $t$ across the vector of indicator variables $\mathbf{z}_{-i}$, ignoring the token that is currently being sampled.

## 2.5 Computing Final Parameter Estimates

We compute final estimates (as well as estimates to be used for log-likelihood computations during inference) of the model parameters as follows:

$$\hat{\theta}_t^{(d)} = \frac{N_{td}^{YD} + \alpha}{\sum_{t'=1}^{T}(N_{t'd}^{YD} + \alpha)} \tag{7}$$

$$\hat{\pi}_r^{(t)} = \frac{N_{rt}^{CT} + \delta}{\sum_{r'=1}^{R}(N_{r't}^{CT} + \delta)} \tag{8}$$

$$\hat{\phi}_w^{(t)} = \frac{N_{wt}^{ZT} + \beta}{\sum_{t'=1}^{T}(N_{w't}^{ZT} + \beta)} \tag{9}$$

The final estimates for the parameters of the spatial distributions are equivalent to estimates used during inference, described previously.

Figure 2: Jensen-Shannon (JS) distances between pairs of topics learned using distinct subsets of training documents. For each topic $t = 1...100$ learned using training Subset 1, we show the JS-distances between the topic $t$ and all topics learned from training Subset 2. The JS-distance between topic $t$ and the topic it was aligned with using a matching algorithm is indicated using a red '×'. Topics from training Subset 1 are sorted in terms of the JS-distance between the topic and its aligned topic from Subset 2.

## 3 Topic Stability Analysis

Given that one goal of our models are to work towards a "functional neuro-anatomical atlas", it is important to consider how stable the topic solutions provided by the model are. That is, if the model is identifying functional regions that are consistent with true underlying neuroanatomical patterns, we expect that these regions should be consistently identified regardless of the specific training data used by the model. To investigate the stability of our topic solutions, we randomly partitioned the Neurosynth database into two equal halves—training Subset 1 and Subset 2—where each of these subsets contained 5,681 complete documents. For each of the two subsets, we trained a "constrained subregions" GC-LDA model using $\gamma = .01$ and all other hyper-parameters equal to those described in Section 3.2 of the main paper.

To evaluate the similarity between the topic solutions identified from the training subsets, we followed a procedure similar to the one described for alignment of standard LDA topics in [6] (although note that in [6] the authors used the same training data but different random initializations to produce two separate topic solutions). Specifically, we computed a $T$-by-$T$ "dissimilarity" matrix, where element $i, j$ of the matrix corresponded to the dissimilarity between the $i$th topic in training Subset 1 and the $j$th topic in Subset 2. We defined the dissimilarity between two topics as the sum of the Jensen-Shannon (JS) distances [3] between the probability distributions over words and the spatial probability distributions for the two topics. Given these dissimilarity matrices, we aligned each topic $t = 1...T$ learned from training Subset 1 with a single topic from training Subset 2, using a greedy algorithm which iterated $T$ times over the following steps: (1) find the lowest remaining dissimilarity value in the dissimilarity matrix, and store the row and column indices as a mapping from the topics in Subset 1 to Subset 2, then (2) remove the corresponding rows and topics from the matrix.

Given the aligned topic sets, we qualitatively evaluated the similarities between the aligned topic pairs in terms of both their spatial and linguistic distributions. Additionally, for each topic $t$ from training Subset 1, we visualized the distribution of JS-distances between its "aligned" topic and all non-aligned topics, as illustrated in Figure 2. From Figure 2, it is clear that for many of the best-aligned topics, the JS-distance between the aligned topics lies outside of the distribution of distances for the non-aligned topics. Based on these analyses, we estimate that approximately 50% of topics identified by the GC-LDA model are stable, and will be consistently extracted, independent of the specific training documents. We note that these analyses are only heuristic in nature, and in future work we hope to formalize a concrete procedure for assessing topic stability.

## Footnotes

[1] An implementation of GC-LDA is available at `http://github.com/timothyrubin/python_gclda`