[Reviews · NeurIPS 2016]

Reviewer 1

Summary

The submission #635, entitled " Generalized Correspondence-LDA Models (GC-LDA) for Identifying Functional Regions in the Brain" presents a topic model that accounts for both word occurrence and associated locations to build a cognitive atlas from the brain imaging literature. The main contribution of the paper is a novel "generalized correspondence model" that is derived into three versions, e.g. including (or not) symmetry constraints on the spatial distributions involved. The model is handled in a Bayesian framework, and the results display an improved predictive accuracy in cross-validation when the full power of the correspondence model is used.

Qualitative Assessment

This paper addresses an important topic, and presents a clear, well-detailed and well-presented framework. I find the results and their illustration compelling. In enjoyed reading the paper. my only major concern regards the cross-validation scheme adopted in the paper (see below). More in detail -------------- 8: 'probability distribution over neural activity' is a bit vague and potentially misleading to describe the kind of statistics made in coordinate-based meta-analysis. 99-111: I find this paragraph a bit awkward. To me the message boils down to: although gamma does not enjoy a very clear interpretation in the generative model, it plays the role of a regularizer. Presenting it in such a way would clarify the presentation. 125 why not 'semantic' instead of 'linguistic' ? Would sound better to me. 130-132: Again, a rather awkward way to introduce symmetry about the x=0 plane. Fig. 2: I did not get the meaning of the colors. Usually, red/blue refer to positive/negative values, but it cannot be the case here. 192 the authors probably mean A2. 238-241: I would _really_ prefer a cross-validation across documents, as it would probably fit with the properties that you expect from a cognitive atlas. The authors should --at least-- explain why they did not do it that way, and if possible, describe the performance of cross-document generalization. The paper did not address model selection (how many classes ?). While I understand that this is ill-posed and potentially impossible in their unsupervised setting, I think that the question should be discussed. Using GMMs with two classes is fairly limited. Why not using more general spatial distributions, such as kernel density estimators ? Gibbs sampling has a strong drawback wrt variational schemes: its computational cost. I have no idea of the computation time for the model (this should be stated at least in SM) , but I think that it would be great to perform a comparison: either confirm the weakness of the variational approach or propose it as an efficient alternative.

Confidence in this Review

3-Expert (read the paper in detail, know the area, quite certain of my opinion)


Reviewer 2

Summary

This paper presents a set of joint spatial and textual topic models, with experiments showing these improve topic quality and the ability to predict held-out data. The model is based on Correspondence-LDA, with increased flexibility.

Qualitative Assessment

Generally I like this paper. The one substantial critique I could make is that evaluating against a neural network approach would strengthen the paper. I don't see that as critical however. Displays like Fig 2 would be less natural from a neural net but the results on held-out likelihood would likely be dramatically better. Beyond that I have a number of smaller comments and suggestions. "lateralization of cognitive functions" should be defined when it's introduced (it becomes clear what this means later) The first reference given for correspondence LDA in the intro should be [2] instead of [3] It would help if you said exactly what the "documents" are earlier. It's not until middle of page 5 that I could tell these were abstracts from papers rather than e.g. words used as stimuli in some fMRI experiment. The graph Fig 1 A seems unnecessary and confusing. Fig B and C are all that are used and A is not really a generalization of B and C because C has the c parameter that has N_X copies rather than T copies. The function f is never really defined in step b2 is never really defined by name, it's replaced by P(x_i | Lambda)

Confidence in this Review

2-Confident (read it all; understood it all reasonably well)


Reviewer 3

Summary

This paper made several modifications of the correspondence LDA model and apply them for identifying functional regions in the brain.

Qualitative Assessment

While this paper made a nice contribution which successfully modifies the correspondence LDA model for brain image modeling, the significance of this contribution is below the NIPS theoreshold. In particular, neither the problem or the method is novel. First, as acknowledged by the authors, the problem -- neuroimaging meta analysis -- was well established and have been widely studied. Second, the proposed method is incremental variant of the correspondence LDA, whose technical value is limited. In light of this, I vote for rejection.

Confidence in this Review

2-Confident (read it all; understood it all reasonably well)


Reviewer 4

Summary

The authors describe an LDA variant capable of extracting anatomical regions based on their function. Function is derived using a database of neuroimaging publications containing abstracts and peak activations. Both spatial extent and cognitive function are parametrized by a probability distribution. They show that assuming subregions is better than using a single multivariate Gaussian. The subregions are created using 2 Gaussian distributions and if these are bilateral performance is further improved slightly.

Qualitative Assessment

The authors show how the performance varies as a function of the hyperparameter gamma but does not show how to actually choose it in practice. I suppose that it is chosen based on the log-likelihood of the test set… Furthermore it would have nice to see the performance on a held out test set using the optimized hyperparameter. It would furthermore be interesting to know how the cognitive mapping changes as a function of gamma. This could provide information about how stabile the solutions are. Figure 2 could use a more descriptive caption. Also how many topics are reliably extracted?

Confidence in this Review

2-Confident (read it all; understood it all reasonably well)


Reviewer 5

Summary

The primary objective of this paper is to tackle the problem of topic modeling in large fMRI datasets and to create a brain region-functional description correspondence employing the latent topics approach. The authors propose an extension of the Correspondence-LDA, where they introduce an additional step to get the word labels: the distribution of words to describe a document does not depend directly on the set of topics used to generate the document/image, but is governed by the intermediate random variable $z$, which allows sampling from the full set of topics. The hyperparameter $\gamma$ allows some flexibility and tradeoff in correspondence between the image topics and the full set of topics. $\gamma$ may be viewed as an additional regularizer which allows previously unobserved word-activation pairs to appear and thus reduces the overfitting which may happen while training the Correspondence-LDA model. This generalized model was equipped with three generative schemes to produce the brain activation pattern with different constraints. Cross-validation experiments indicate the usefulness of the smoothed intermediate word topic distribution $z$ in certain range of the $\gamma$ hyperparameter.

Qualitative Assessment

In general, the paper is not a giant leap in the large-scale meta-analysis of fMRI studies, but definitely a small step worth considering. The extracted latent topics are interpretable and the brain regions activations are also meaningful. This should not come as surprise, as with $\gamma=0$ the model reduces to the Corr-LDA. The proposed constraints on the brain activation part of the model - subregions and symmetric subregions - are greatly inspired by the existing accumulated knowledge of the pattern of activations, and may be useful in post-hoc meta-analysis of the past studies, but I would not recommend applying the symmetry constraint for the future single-study researches, as it is definitely very biased, and, as the authors report, may be used to estimate the functional asymmetry. The paper is well written and has a nice illustration how the proposed models differ in brain regions identification. In the introduction section the authors claim that the "primary goal of cognitive neuroscience is to find a mapping from neural activity onto cognitive processes", which sounds as an overstatement. One obviously should not diminish the other directions of research in this area, the focus is mainly on the cognitive processes studies. Typos found: - line 16, "goal of cognitive neuroscience it" -> "goal of cognitive neuroscience is" - line 287 "features GC-LDA" -> "features of GC-LDA"

Confidence in this Review

2-Confident (read it all; understood it all reasonably well)